# Towards Scalable Language-Image Pre-training for 3D Medical Imaging

**Chenhui Zhao**                                           *chuizhao@umich.edu*
*University of Michigan*

**Yiwei Lyu**                                              *yiweilyu@umich.edu*
*University of Michigan*

**Asadur Chowdury**                                        *achowdur@umich.edu*
*University of Michigan*

**Edward Harake**                                          *eharakej@umich.edu*
*University of Michigan*

**Akhil Kondepudi**                                        *akhilk@umich.edu*
*University of Michigan*

**Akshay Rao**                                             *akshayro@umich.edu*
*University of Michigan*

**Xinhai Hou**                                             *xinhaih@umich.edu*
*University of Michigan*

**Honglak Lee**                                            *honglak@umich.edu*
*University of Michigan*

**Todd Hollon**                                            *tocho@umich.edu*
*University of Michigan*

**Reviewed on OpenReview:** *https://openreview.net/forum?id=WxHf4EcBWA*

## Abstract

The scalability of current language-image pre-training for 3D medical imaging, such as CT and MRI, is constrained by the need for radiologists to manually curate raw clinical studies. In this work, we pioneer pre-training directly on uncurated studies, which both aligns more closely with the clinical workflow and provides a natural path to scalability. However, the unique structure of such data presents new challenges for existing model architectures, which were originally designed for 2D slices or single 3D scans. To address this, we introduce a novel hierarchical attention mechanism inspired by the intrinsic hierarchy of radiology data: slice, scan, and study. We denote our framework as Hierarchical attention for Language-Image Pre-training (HLIP). Trained on 220K studies with 3.13 million scans for brain MRI and 240K studies with 1.44 million scans for head CT, HLIP achieves state-of-the-art performance, *e.g.*, +10.5% balanced ACC on the proposed publicly available brain MRI benchmark Pub-Brain-5; +8.3% and +1.7% macro AUC on head CT benchmarks CQ500 and RSNA, respectively. HLIP also exhibits strong generalizability on existing 3D medical language-image pre-training benchmarks, *e.g.*, +4.3% macro AUC on the Rad-ChestCT benchmark when pre-trained on CT-RATE. These results demonstrate that, with HLIP, directly pre-training on uncurated clinical datasets is a scalable and effective direction for language-image pre-training in 3D medical imaging. The code is available at https://github.com/zch0414/hlip

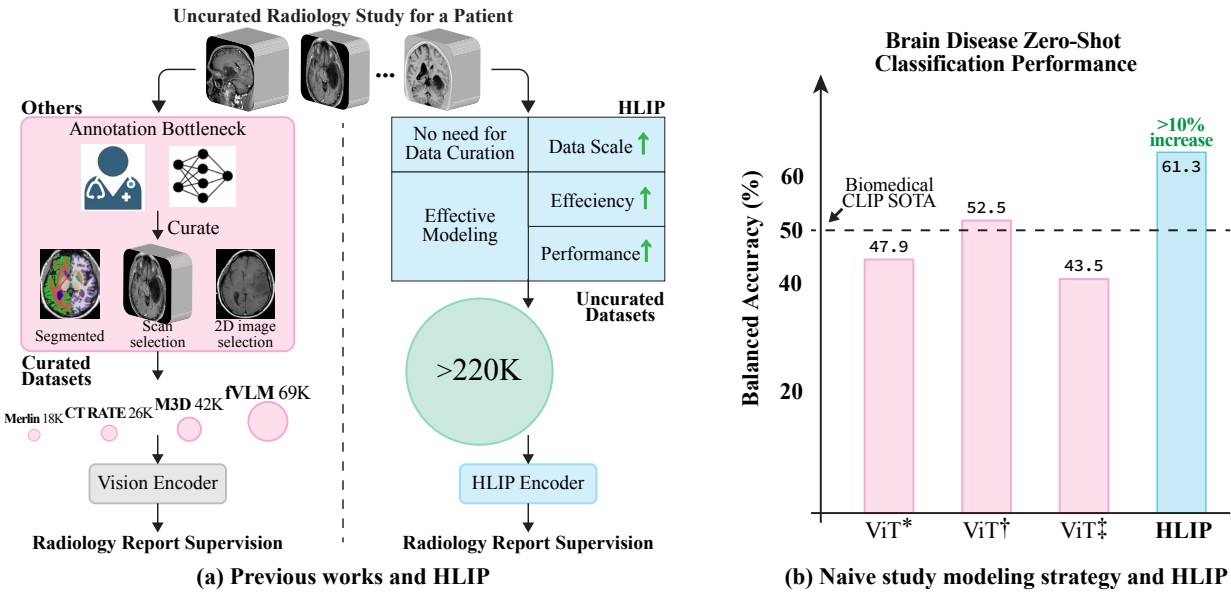

Figure 1: Illustration of (**a**) an uncurated study for a patient. While previous work has relied on annotation and curation, HLIP enables language-image pre-training directly on uncurated data. (**b**) Despite training on large-scale domain-specific datasets, naively modeling the uncurated study with a vanilla ViT, *e.g.*, by randomly selecting a scan at each training step*, encoding scans independently before study aggregation†, or directly encoding the entire study‡, yields performance only comparable to the SOTA trained on PubMed corpus, whereas HLIP outperforms these by a large margin.

# 1 Introduction

Language-supervised pre-training is well-suited for radiology, where each study comprises medical images paired with a corresponding radiologist's report. This natural alignment between visual and textual information has motivated the adaptation of language-image pre-training methods, such as CLIP (Zhang et al., 2022a; Radford et al., 2021), to learn clinically meaningful radiology representations. CLIP-based models are especially notable for their clinical utility: they demonstrate strong zero-shot transfer performance on diagnostic tasks (Hamamci et al., 2024; Blankemeier et al., 2024), and their encoders consistently improve performance on multimodal learning benchmarks (Bai et al., 2024; Shui et al., 2025). In the domain of 2D medical imaging, such as chest X-rays, language-supervised pre-training is a key driver for integrating computer vision into clinical workflows (Boecking et al., 2022; Wang et al., 2022b; Tiu et al., 2022).

However, language-image pre-training in 3D medical imaging has yet to reach the scale or performance demonstrated in 2D modalities. For instance, chest X-ray CLIP models have been trained on a 500K corpus (Wang et al., 2022b; Tiu et al., 2022; You et al., 2023), achieving human-level performance on multiple diagnostic tasks (Tiu et al., 2022). In contrast, progress in 3D medical imaging remains limited, with existing models underperforming relative to their 2D counterparts (Hamamci et al., 2024; Bai et al., 2024). We attribute the performance this gap to two primary factors: data annotation that constrains the training scale, and architectural limitations arising from the complexity of 3D medical imaging.

Computed tomography (CT) and magnetic resonance imaging (MRI) generate 3D volumetric images across various anatomical regions, including the brain, chest, and abdomen. As illustrated in Figure 1, a standard MRI study typically includes several sequences (e.g., T1-weighted, T2-weighted, and FLAIR), each contributing distinct diagnostic information. Similarly, CT studies often include scans acquired with varying orientations or scanner settings within the same study. To perform language-image pre-training for such data, a common strategy is to curate datasets by having radiologists manually select a representative scan or slice from each study, as shown in Figure 1(a) (Blankemeier et al., 2024; Hamamci et al., 2024; Bai et al., 2024; Shui et al., 2025), which presents a significant barrier to the scalability. In contrast, pre-training on uncurated studies

aligns more closely with real-world practice and readily expands the data scale, as it imposes no additional burden on radiologists. Despite this, the unique structure of such data presents new challenges for current visual encoders, which were originally designed for 2D images or single 3D scans (Dosovitskiy et al., 2020; Liu et al., 2021; Ryali et al., 2023). As shown in Figure 1(b), even trained on a large-scale domain-specific dataset, naively encoding uncurated studies with the Vision Transformer (ViT) (Dosovitskiy et al., 2020) results in only comparable zero-shot transferability to the state-of-the-art (SOTA) biomedical CLIP (Nie et al., 2025) trained on the PubMed corpus. Particularly, encoding the entire study can produce tokens on the order of $10^4$, which both incurs substantial computational overhead and limits performance (Barbero et al., 2024).

In this work, we first address the key barriers to scaling language-image pre-training for 3D medical imaging by pioneering the use of uncurated studies. Second, to effectively extract features from such data, we introduce a novel hierarchical attention mechanism leverages the natural hierarchy of radiology data: slice, scan, and study. We name this framework _Hierarchical attention for Language-Image Pre-training_ (HLIP). Unlike architectural designs such as Swin (Liu et al., 2021), MViT (Fan et al., 2021; Li et al., 2022b), and Hiera (Ryali et al., 2023), HLIP leverages the inherent data structure to define the attention scope, making it suitable for uncurated studies that contain multiple 3D scans. Compared to window attention that captures only local features, slice or scan attention can capture all diagnostic features while also providing constructive priors for learning study representations. Moreover, slice and scan attention is already much lighter than study attention, and such minimal adaptation of the original ViT remains orthogonal to flash attention (Dao et al., 2022) and patch dropout (Li et al., 2023), further reducing the computational burden.

Trained on our health system, HLIP outperforms the SOTA (Nie et al., 2025) on the proposed publicly available brain MRI benchmark, Pub-Brain-5, by 10.5% balanced ACC; and surpasses the head CT foundation model (Zhu et al., 2025) by 8.3% and 1.7% macro AUC on the CQ500 (Chilamkurthy et al., 2018) and RSNA (Flanders et al., 2020) benchmarks, respectively. HLIP also demonstrates strong generalizability on the curated 3D medical language-image pre-training benchmark CT-RATE (Hamamci et al., 2024), which contains only one scan per study, outperforming the SOTA (Liu et al., 2023) by 4.3% macro AUC on the external evaluation Rad-ChestCT (Draelos et al., 2021). Our paper makes the following contributions:

- We introduce HLIP, an effective and scalable language-image pre-training framework for uncurated 3D medical imaging, that leverages a novel hierarchical attention mechanism derived from the natural structure of radiology data.

- We conduct the largest-scale training for 3D medical imaging to date, using 220K studies with 3.13 million scans for brain MRI and 240K studies with 1.44 million scans for head CT.

- We demonstrate the state-of-the-art performance on multiple benchmarks spanning diverse modalities and anatomical regions, including brain MRI, head CT and chest CT.

- We release the following assets to the public: a brain MRI benchmark for zero-shot classification, an effective language-image pre-training implementation for 3D medical imaging, the pre-training recipe, and model checkpoints.

## 2 Related Work

### 2.1 Language-Image Pre-training in Radiology

has been developed for 2D and 3D medical imaging, facilitated by public datasets such as CheXpert (Irvin et al., 2019), MIMIC-CXR (Johnson et al., 2019), and CT-RATE (Hamamci et al., 2024). In 2D imaging, techniques such as local alignment (Huang et al., 2021; Wang et al., 2022a; Müller et al., 2022), knowledge enhancement (Wang et al., 2022b; Wu et al., 2023), and longitudinal analysis (Bannur et al., 2023) have been explored. CheXZero (Tiu et al., 2022) demonstrates strong empirical results comparable to those of human experts in the domain of chest X-rays. BiomedCLIP (Zhang et al., 2023a) and ConceptCLIP (Nie et al., 2025) have been trained on more than _15 million_ sample pairs, achieving strong performance across modalities, including radiology. In addition, other works (Boecking et al., 2022; You et al., 2023; Zhang et al., 2023b) have also contributed significant insights and achieved impressive performance.

3D medical imaging offers a more comprehensive view of anatomical structures. However, due to its computational cost and limited data availability, several studies (Cao et al., 2024; Wang et al., 2024; He et al., 2024) focus on bridging the domain gap between 2D and 3D. For example, UniMedI (He et al., 2024) learns a shared feature space for both 2D and 3D modalities and demonstrates improvements across both. BIUD (Cao et al., 2024) distills 3D representations from a well-trained 2D model (Tiu et al., 2022), significantly improving data efficiency.

More recently, aided by advances in hardware and the availability of public datasets (Hamamci et al., 2024), several studies (Hamamci et al., 2024; Bai et al., 2024; Shui et al., 2025; Lai et al., 2025) have performed language-image pre-training on 3D imaging, surpassing methods (Cao et al., 2024) that still rely on 2D representations. Specifically, CT-CLIP (Hamamci et al., 2024) has been pre-trained on 20,000 paired chest CT scans and reports. M3D (Bai et al., 2024) explores a versatile multi-modal large language model designed for universal 3D medical imaging analysis. fVLM (Shui et al., 2025) proposes a fine-grained pre-training framework that relies on segmentation to perform organ alignment. However, while these methods have demonstrated promising performance on various tasks such as zero-shot abnormality detection and report generation, several factors limit their training scalability and real-world applicability. For example, CT-CLIP and M3D are constrained to curated studies containing only a single imaging scan. Such studies rely heavily on human annotation and also fail to reflect real-world scenarios, where studies typically include multiple scans. fVLM further depends on a segmentation model, which introduces bias from segmentation quality and largely limits its scalability and generalizability. Moreover, these methods suffer from inefficient modeling of 3D medical imaging, leading to small batch sizes (*e.g., 48*), which are insufficient for effective language-image pre-training (Radford et al., 2021).

## 2.2 Efficient Language-Image Pre-training

is crucial, as it directly affects the number of sample pairs seen during training and contrasted per batch, two key factors of model capacity. Existing hierarchical architectures (Liu et al., 2021; Hamamci et al., 2024) have been widely adopted to improve efficiency. However, as analyzed in Appendix E.2, these models may be less efficient than commonly assumed in the context of 3D medical imaging, even when compared with the original ViT. From the architecture perspective, components such as relative position embedding are expensive for 3D inputs and is not compatible with recent advancements like flash attention (Dao et al., 2022). From the training perspective, FLIP (Li et al., 2023) demonstrates a favorable trade-off by randomly removing 50% of tokens during training. However, models that require a fixed activation shape (Liu et al., 2021; Li et al., 2022b;a; Ryali et al., 2023) cannot benefit from this strategy. Moreover, many works in radiology (Matsoukas et al., 2022; Tiu et al., 2022; Zhao & Shen, 2024; Shui et al., 2025) have demonstrated the benefits of the universal feature learned by MAE (He et al., 2022). Therefore, modeling 3D medical imaging with minimal adaptation of the original ViT (Dosovitskiy et al., 2020) is a promising research direction and stands to benefit from recent advancements such as flash attention, patch dropout, and pre-trained models.

## 3 Method

Our goal is to perform language-image pre-training on uncurated studies. Given a study $S \in \mathbb{R}^{M \times 1 \times D \times H \times W}$, where $M$ denotes the number of single-channel 3D scans, and $D$, $H$, and $W$ represent the depth, height, and width dimensions of each scan, respectively. Following the ViT (Dosovitskiy et al., 2020), each scan is divided into a grid of non-overlapping volumes of size $(\frac{D}{d}, \frac{H}{h}, \frac{W}{w})$. These volumes are projected into visual tokens $F_s \in \mathbb{R}^{N \times c}$, where $N = M \times d \times h \times w$ is the total number of tokens and $c$ denotes the number of channels. As $N$ can be on the order of $10^4$, computing self-attention over all tokens throughout the ViT backbone is prohibitive in memory and also limits performance (Barbero et al., 2024). We introduce a novel hierarchical attention mechanism guided by the inherent data hierarchy of uncurated studies, enabling lighter self-attention while introducing effective priors derived from this hierarchy: slice, scan, and study.

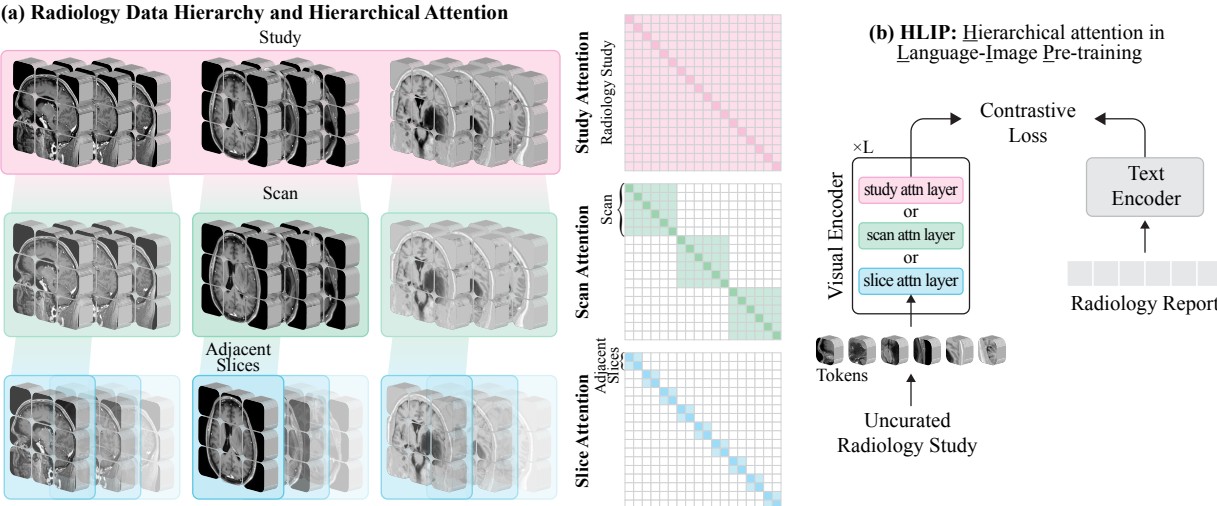

Figure 2: Illustration of **(a)** the radiology data hierarchy for a single patient, including the study, single scan, and adjacent slices. Our hierarchical attention mechanism mirrors this hierarchy and computes self-attention independently within each level. **(b)** Our HLIP framework incorporates a visual encoder that performs attention at different levels. In practice, lightweight slice or scan attention with a few study attention layers suffices to extract features from the full study.

## 3.1 Hierarchical Attention Mechanism

The radiology data exhibits an inherent hierarchical structure for each patient, and our proposed hierarchical attention mechanism mirrors this structure. As illustrated in Figure 2, the data for an individual patient comprises three levels:

- The **study** contains $M$ imaging scans, whose modalities and acquisition planes are selected by a radiologist based on the clinical context. Collectively, these $M$ scans contain all the visual information required for radiologic diagnosis.

- The **scan** contains $D$ slices. While a single scan conveys only partial context of the full study, it still captures the complete extent of the target pathology.

- The **adjacent slices** contains $\frac{D}{d}$ consecutive slices within a single scan. Although it conveys only partial information about the target pathology, the slices can capture focal diagnostic features.

We explore a simple hierarchical attention mechanism grounded in this structure. Specifically, as shown in Figure 2, we compute self-attention independently within each hierarchical level:

- **Study attention** computes a single self-attention operation over all $N$ tokens in a study. The I/O complexity (Aggarwal & Vitter, 1988) of the study attention is $\Omega(N^2 + N \times c)$ (Dao et al., 2022), as detailed in Appendix E.

- **Scan attention** computes $M$ independent self-attention operations, each over $d \times h \times w$ tokens within a single scan. The I/O complexity of the scan attention is $\Omega(\frac{N^2}{M} + N \times c)$.

- **Slice attention** computes $M \times d$ independent self-attention operations, each over $h \times w$ tokens within a group of adjacent slices. The I/O complexity of the slice attention is $\Omega(\frac{N^2}{M \times d} + N \times c)$.

The proposed hierarchical attention mechanism contrasts with existing methods that modify attention computation using convolution (Fan et al., 2021; Li et al., 2022b), pooling (Ryali et al., 2023), or shifted windows (Liu et al., 2021), all of which require either regular activation shapes or attention masks. Our mechanism relies solely on a simple reshape operation, as there is no overlap between different scans or groups of adjacent slices[1], allowing greater flexibility for uncurated studies that comprise multiple scans. As shown in Figure 2, each layer in the ViT can perform attention at any level. In practice, we evenly divide the ViT backbone into four subsets of layers (e.g., three layers per subset for the 12-layer ViT-B) and apply study attention only to the last layer of each subset, while the remaining layers perform the lighter scan or slice attention. Moreover, this intentionally simple design also enables seamless integration with recent efficiency advances such as patch dropout (Li et al., 2023) and flash attention (Dao et al., 2022).

### 3.2 Implementation

**Model.** Our visual encoder is a MAE pre-trained ViT-B (He et al., 2022). The input is rescaled to [0,1] and normalized with MAE's averaged *mean* and *std*. Compared to the original ViT designed for 2D RGB images, our vision encoder only has three differences: (1) each scan is divided into 3D volumes instead of 2D patches; (2) the positional embedding encodes 3D spatial coordinates within each scan, along with an additional dimension to distinguish different scans; The *cls token* is minimally adapted to propagate information across layers that perform different hierarchies of attention.

For (1), which employs a convolution layer (He et al., 2022), we adopt average inflation initialization (Zhang et al., 2022b). Specifically, we first sum the 2D weights along the channel dimension to accommodate a single-channel input, then replicate them $\frac{D}{d}$ times to construct the 3D weights, and finally scale by a factor of $\frac{d}{D}$. We set the token size as (8,16,16) by default. For (2), we *tile* the pre-trained 2D positional embedding $P_{mae} \in \mathbb{R}^{h \times w \times c}$ with a 1D sinusoidal positional embedding for slices, $P_{slice} \in \mathbb{R}^{d \times c}$, and another for scans, $P_{scan} \in \mathbb{R}^{M_{max} \times c}$. $M_{max}$ indicates the maximum number of scans our model can process without interpolation. During training, to construct a batch, we randomly sample $M$ scans along with $M$ positional embeddings from $P_{scan}$, which forms the final positional embedding $P \in \mathbb{R}^{M \times d \times h \times w \times c}$. We set $\mathtt{M_{max}}$=40 and M=10 by default. For (3), the *cls token* may lose gradient continuity across different hierarchies. To address this issue while maintaining the efficiency of our architecture, we propagate the *cls token* using a combination of cloning and averaging. For example, when transitioning from the study to scan attention layer, we distribute the *cls token* across $M$ scans by cloning. In the reverse direction, we aggregate information from $M$ *cls tokens* into a single *cls token* by averaging them. The transition between other hierarchies (*e.g.*, study and slice or scan and slice) follows the same procedure. Empirically, we find this strategy to be sufficient, performing better than alternatives such as weighted average or global pooling at the end of the encoder.

For uncurated datasets, we use scan attention with *4 evenly* distributed study attentions. A curated dataset with only one scan per study corresponds to the special case of M=1 in our scenario. For such datasets, we employ slice attention with *4 evenly* distributed scan attentions; We use PubMedBERT (Gu et al., 2020) as our text encoder. Model configurations are provided in Appendix B.

**Datasets.** Given that HLIP enables pre-training on real-world, clinical studies, we collect two datasets within our health system, namely *BrainMRI220K* and *HeadCT240K*. BrainMRI220K contains 220,993 MRI studies. We hold out 992 studies for hyperparameter tuning using the retrieval task. In the training split, the number of scans per study ranges from 1 to 162, with the third quartile at 17; the number of slices per scan ranges from 5 to 500, with the third quartile at 80. HeadCT240K contains 244,253 CT studies for training and 998 held-out studies. The number of scans per study ranges fro 1 to 71, with the third quartile at 6; the number of slices per scan ranges from 5 to 500, with the third quartile at 110. More details about these two datasets are provided in Appendix A.

---

[1]With the original attention layer unchanged, study attention takes the input of size $(B, M \times d \times h \times w, c)$; scan attention takes the input of size $(B \times M, d \times h \times w, c)$; and slice attention takes the input of size $(B \times M \times d, h \times w, c)$, where $B$ denotes the batch size.

**Preprocessing.** We do not standardize the orientation or spacing. Instead, we consistently align the depth dimension with the through-plane axis of each scan, and then resize it to a fixed shape of `(48,224,224)`. As this differs from prior practice (Blankemeier et al., 2024; Hamamci et al., 2024; Shui et al., 2025), we provide a discussion in Appendix A.3. For brain MRI, we apply `[0.5,99.5]` percentile clipping to the intensity values. For head CT, we expand each scan into three separate scans, with Hounsfield Unit (HU) values truncated to `[0,120]` for soft tissue, `[-20,180]` for blood vessels, and `[-800,2000]` for bone.

**Pre-training.** Our implementation builds upon OpenCLIP (Cherti et al., 2023) and FLIP (Li et al., 2023). We apply 25% patch dropout by default as regularization and acceleration. We do not apply additional augmentation or unmasked fine-tuning. Training 20 epochs on our uncurated datasets takes ~1 day with a batch size of 256 on 8 L40 GPUs. More details are provided in Appendix B.2.

## 4 Experiments

We apply HLIP to brain MRI, head CT, and chest CT. For brain MRI, we construct a new benchmark for zero-shot classification, using public datasets (Dufumier et al., 2022; Liu et al., 2023; Baid et al., 2021; LaBella et al., 2023; Moawad et al., 2024; Kazerooni et al., 2024; Rudie et al., 2024; Link et al., 2024). For head CT and chest CT, we follow evaluation protocols from previous work (Hamamci et al., 2024; Shui et al., 2025; Zhu et al., 2025). Experiments on our uncurated brain MRI and head CT datasets underscore the importance of effective modeling and demonstrate superior performance of HLIP over current foundation models (Zhang et al., 2023a; Nie et al., 2025; Zhu et al., 2025; Yang et al., 2024). Experiments on curated chest CT datasets (Draelos et al., 2021; Hamamci et al., 2024) further isolate the effectiveness of the proposed hierarchical attention mechanism. Finally, we conduct comprehensive ablation and analysis.

### 4.1 Brain MRI

**Pub-Brain-5.** To the best of our knowledge, no publicly available benchmark exists for zero-shot transfer evaluation on brain MRI tasks[2]. To this end, we construct a benchmark, named *Pub-Brain-5*, based on existing publicly available brain MRI datasets. Pub-Brain-5 comprises 18,343 studies drawn from Open-BHB (Dufumier et al., 2022), the Stroke dataset (Liu et al., 2023), BraTS 2023 (Baid et al., 2021; LaBella et al., 2023; Moawad et al., 2024; Kazerooni et al., 2024), NYU-Mets (Link et al., 2024), and UCSF-Mets (Rudie et al., 2024). It spans five classes: healthy (3,984), acute stroke (2,871), glioma (1,614), meningioma (1,141), and metastasis (8,733). For each study, the number of scans ranges from 1 to 14. We also construct a subset of Pub-Brain-5, namely *Pub-Brain-5-GT*, which contains lesion-containing slice annotations derived from the original segmentation ground truth. Pub-Brain-5-GT comprises 8,944 studies covering the same five classes: healthy (3,984), acute stroke (2,372), glioma (1,350), meningioma (1,000), and metastasis (238). We evaluate three zero-shot tasks: (i) binary anomaly detection (*i.e.*, distinguishing pathological from healthy studies); (ii) three-way tumor classification; and (iii) five-way disease classification.

**Baselines.** We first evaluate the zero-shot transferability of two biomedical CLIP models on our benchmark: BiomedCLIP (Zhang et al., 2023a) and ConceptCLIP (Nie et al., 2025). Since both models require 2D inputs, we generate study predictions by applying *average pooling* to the outputs across all slices. We acknowledge that slices without lesions introduce noise for these 2D baselines; however, this is an intrinsic limitation of such models. We support these two baselines on Pub-Brain-5-GT, where predictions are made only on lesion-containing slices, while HLIP continues to take the raw study as input. In addition, we evaluate Prima (Lyu et al., 2026), which represents the *vqvae + hierarchical transformer* architecture family (Hamamci et al., 2024; Lyu et al., 2026), as well as a vanilla ViT that encodes the entire study.

**Implementation Details.** We use the prompt "*This brain MRI shows: {disease}.*" to perform zero-shot inference (Radford et al., 2021) with biomedical CLIP models. HLIP and the vanilla ViT are trained on BrainMRI220K as described in Section 3.2, while Prima (Lyu et al., 2026) is re-trained on the same dataset using the training recipe provided in the original paper. These three models employ the prompt "*This MRI study shows: {disease.}*" during zero-shot transfer.

---

[2]BrainMD (Wang et al., 2024) is not publicly available due to an unexpected privacy policy.

Table 1: Results of zero-shot classification on Pub-Brain-5 and Pub-Brain-5-GT. We report balanced accuracy, with the best results highlighted in **bold** and the second-best result highlighted in underline. "*+annotation*" indicates only predicting on lesion-containing slices which are manually annotated.

| Method | Anomaly Detection | | | | | Tumor | Disease |
|---|---|---|---|---|---|---|---|
| | Stroke | Glioma | Meningioma | Metastasis | *mean* | | |
| *Pub-Brain-5* | | | | | | | |
| BiomedCLIP (Zhang et al., 2023a) | 64.7 | 87.8 | 63.6 | 59.8 | 69.0 | 50.4 | 31.5 |
| ConceptCLIP (Nie et al., 2025) | 66.8 | 91.9 | 57.7 | 67.9 | 71.1 | 35.7 | 30.9 |
| Prima (Lyu et al., 2026) | 78.8 | 89.3 | 70.8 | 64.7 | 75.9 | 42.8 | 31.6 |
| ViT (our impl) | 72.8 | 93.4 | 72.9 | 63.1 | 75.6 | 45.7 | 43.4 |
| HLIP (ours) | 91.5 | 89.2 | 79.2 | 78.1 | **84.5** | **63.3** | **63.9** |
| *Pub-Brain-5-GT* | | | | | | | |
| BiomedCLIP | 66.7 | 88.2 | 63.8 | 74.6 | 73.3 | 46.2 | 33.1 |
| *+annotation* | 86.4 | 94.1 | 75.8 | 75.0 | 82.8 | 45.7 | 45.3 |
| ConceptCLIP | 69.6 | 92.1 | 57.8 | 69.5 | 72.3 | 35.2 | 31.6 |
| *+annotation* | 93.6 | 97.8 | 70.8 | 76.8 | **84.8** | 39.4 | 50.8 |
| Prima | 61.2 | 81.0 | 87.7 | 53.4 | 70.8 | 45.9 | 31.4 |
| ViT (our impl) | 76.7 | 93.5 | 58.2 | 58.2 | 71.7 | 42.1 | 43.5 |
| HLIP (ours) | 95.0 | 89.2 | 79.6 | 73.4 | 84.3 | **54.8** | **61.3** |

**Results.** Given both benchmarks are imbalanced, we report balanced accuracy (ACC) in Table 1. On Pub-Brain-5, HLIP demonstrates superior performance over all baseline models, outperforming the second-best baseline by 20.5% ACC in disease classification. On Pub-Brain-5-GT, we observe a substantial performance boost for two biomedical CLIP models (Zhang et al., 2023a; Nie et al., 2025) when predictions are restricted to lesion-containing slices, underscoring the fairness of our evaluation for these models even on uncurated studies. Notably, in disease classification, ConceptCLIP (Nie et al., 2025) achieves 50.8% ACC, outperforming Prima (Lyu et al., 2026) and the vanilla ViT trained on our large-scale domain-specific datasets, indicating that merely scaling up data is not a sufficient solution for language-image pre-training in 3D medical imaging. On the other hand, HLIP achieves the new SOTA 61.3% ACC, attributable to the effectiveness of the proposed hierarchical attention mechanism.

HLIP can occasionally lag behind BiomedCLIP (Zhang et al., 2023a) and ConceptCLIP (Nie et al., 2025), particularly in the detection of brain glioma. However, we note that datasets like BraTS (Baid et al., 2021) are likely included in their pre-training corpora, as both models are trained on figure-caption pairs from PubMed. HLIP, on the other hand, does not encounter any instances from public datasets during training. The skull-stripping used in BraTS actually exacerbates the distribution shift. Moreover, HLIP makes predictions directly from the original radiology study, offering greater flexibility for real-world applications.

## 4.2 Head CT

**Implementation Details.** We apply HLIP to HeadCT240K, as described in Section 3.2. For linear-probe evaluation, we benchmark HLIP on CQ500 (Chilamkurthy et al., 2018) and RSNA (Flanders et al., 2020), exactly following the train–test split and hyperparameters of FM-HeadCT (Zhu et al., 2025). For zero-shot evaluation, we evaluate HLIP on the full CQ500 and RSNA datasets and compare it with the vanilla ViT also trained on the same HeadCT240K dataset[3]. We employ the prompt "*This CT study shows: {disease}.*" during zero-shot transfer.

---

[3] Recent CLIP models for 3D medical imaging, such as Merlin (Blankemeier et al., 2024) and M3D (Bai et al., 2024), are excluded from our baselines, as they perform no better than random guessing in zero-shot evaluation on CQ500 and RSNA.

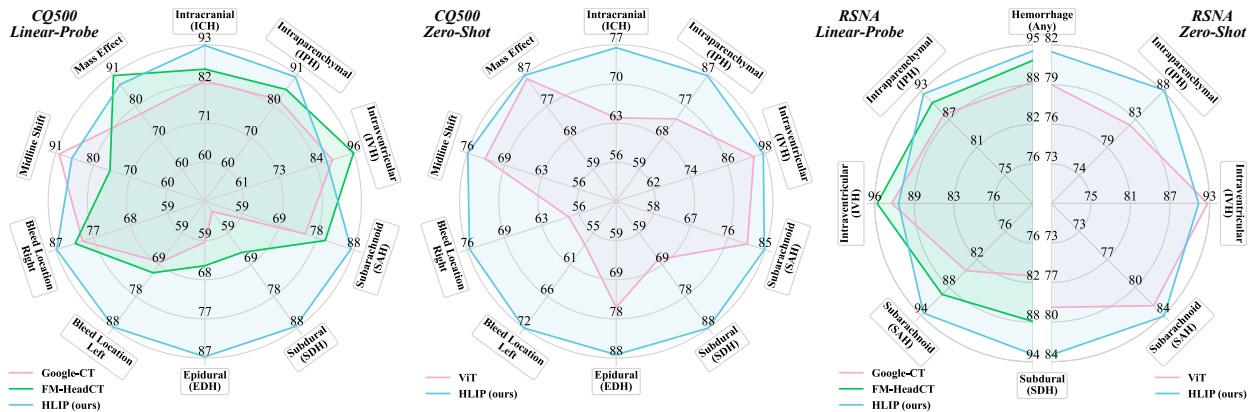

Figure 3: Results of linear-probe and zero-shot evaluations on the CQ500 (Chilamkurthy et al., 2018) and RSNA (Flanders et al., 2020) datasets. We report AUC for each class. In linear-probe, red represents Google CT (Yang et al., 2024); green represents FM-HeadCT (Zhu et al., 2025); and blue represents our HLIP. In zero-shot evaluation, red represents the vanilla ViT and blue represents our HLIP.

**Results.** Consistent with FM-HeadCT (Zhu et al., 2025), we report the area under the ROC curve (AUC) for each class in Figure 3. In linear-probe evaluation, HLIP demonstrates superior performance over FM-HeadCT (Zhu et al., 2025), which is pre-trained on 361,663 studies using DINOv2 (Oquab et al., 2023), and the Google-CT (Yang et al., 2024) foundation model. Specifically, on the CQ500 dataset (Chilamkurthy et al., 2018), HLIP outperforms both FM-HeadCT and Google-CT with macro AUC improvements of 8.3% and 12.2%, respectively; on the RSNA dataset (Flanders et al., 2020), HLIP achieves macro AUC improvements of 1.7% and 5.8% over the these baselines, respectively. Outperforming current foundation models demonstrates the clinical significance of HLIP. In zero-shot evaluation, HLIP consistently outperforms the vanilla ViT, achieving macro AUC improvements of 10.0% on CQ500 and 2.4% on RSNA, further demonstrating the effectiveness of the proposed hierarchical attention mechanism.

## 4.3 Chest CT

**Implementation Details.** Following CT-CLIP (Hamamci et al., 2024) and fVLM (Shui et al., 2025), we apply HLIP to the CT-RATE (Hamamci et al., 2024) training set, then perform internal validation on its test split and external validation on the full Rad-ChestCT dataset (Draelos et al., 2021). The implementation details differ from those in Section 3.2. Following prior works (Cao et al., 2024; Shui et al., 2025), we standardize each scan to a spacing of (3mm,1mm,1mm), truncate HU values to [-1150,350], construct mini-batches with a center crop of size (112,336,336), and apply a token size of (8,24,24). We also use CXR-BERT (Boecking et al., 2022) as the text encoder. Other than the aforementioned modifications, all remaining settings are exactly the same as those described in Section 3.2. HLIP is trained for 20 epochs without patch dropout. Using a batch size of 512 on 4 A40 GPUs, training completes within 6 hours, significantly faster than prior works (Hamamci et al., 2024; Shui et al., 2025), which require several days even with more advanced computational resources. More details are provided in Appendix B.1.

**Results.** Consistent with fVLM (Shui et al., 2025), we report the area under ROC curve (AUC); balanced accuracy (ACC); weighted F1-score (F1); recall; and precision for multi-label zero-shot classification in Table 2. When pre-trained with original reports, HLIP outperforms second-best models by 4.9% AUC on internal and 7.9% AUC on external validation. When pre-trained with reports summarized by large language models (e.g., Qwen (Bai et al., 2023), as used by fVLM), HLIP outperforms fVLM by 0.9% and 3.7% AUC on internal and external validation, respectively. These results demonstrate that, compared with other components such as the VQ-VAE (Van Den Oord et al., 2017) tokenizer (used by CT-CLIP) and anatomical-guided fine-grained alignment (used by fVLM), the proposed hierarchical attention is a more effective and generalizable adaptation for language-image pre-training in 3D medical imaging.

Table 2: Results of multi-label zero-shot evaluation on the CT-RATE (Hamamci et al., 2024) test split for internal validation and the Rad-ChestCT (Draelos et al., 2021) for external validation. The best results are highlighted in **bold**.

| Method | Internal Validation (CT-RATE) | | | | | External Validation (Rad-ChestCT) | | | | |
|---|---|---|---|---|---|---|---|---|---|---|
| | AUC | ACC | F1 | Precision | Recall | AUC | ACC | F1 | Precision | Recall |
| *Supervised by Original Reports* | | | | | | | | | | |
| CT-CLIP (Hamamci et al., 2024) | 73.3 | 66.9 | 70.8 | 32.6 | - - | 63.3 | 59.9 | 64.7 | 34.1 | - - |
| BIUD (Cao et al., 2024) | 71.3 | 68.1 | 71.6 | 33.8 | 67.3 | 62.9 | 60.6 | 65.2 | 33.7 | 59.6 |
| Merlin (Blankemeier et al., 2024) | 72.8 | 67.2 | 70.9 | 33.7 | 70.1 | 64.4 | 61.9 | 66.3 | 34.8 | 61.0 |
| HLIP (ours) | **77.7** | **71.4** | **74.7** | **37.9** | **73.0** | **72.3** | **68.4** | **72.1** | **40.4** | **66.7** |
| *Supervised by Qwen (Bai et al., 2023) Summarized Reports* | | | | | | | | | | |
| fVLM (Shui et al., 2025) | 77.8 | 71.8 | 75.1 | 37.9 | 72.8 | 68.0 | 64.7 | 68.8 | 37.4 | 64.6 |
| HLIP (ours) | **78.7** | **72.4** | **75.5** | **38.4** | **74.1** | **71.7** | **67.7** | **71.4** | **39.8** | **66.9** |

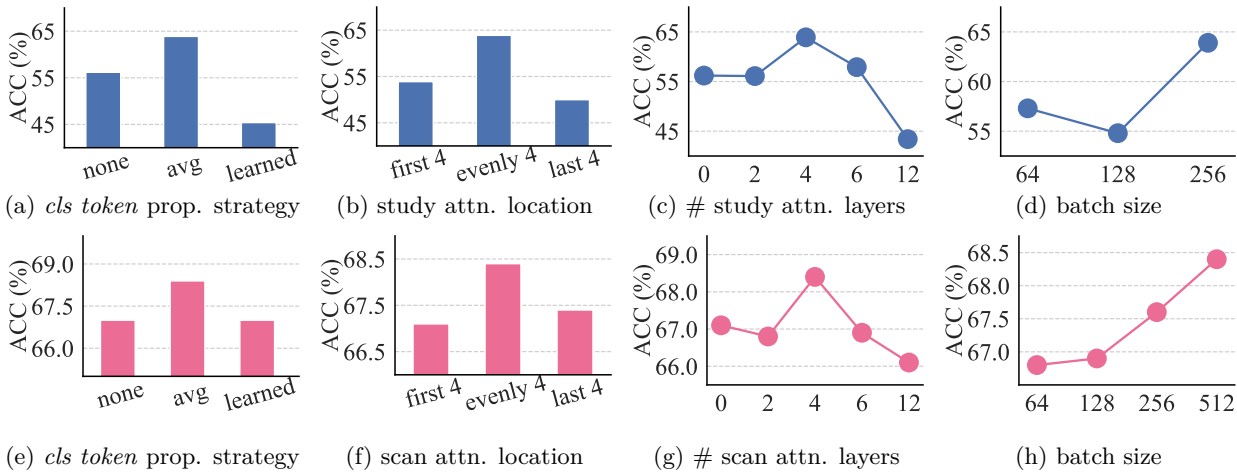

Figure 4: Ablation study on the Pub-Brain-5 (a)-(d) and Rad-ChestCT (Draelos et al., 2021) dataset (e)-(h). We report balanced accuracy (ACC) on both datasets.

## 4.4 Ablation Study

In Figure 4, we present an ablation study on the key components of our visual encoder, including the *cls token* propagating strategy (Figure 4a and 4e); the location of dense attention (Figure 4b and 4f); and the number of dense attention layers (Figure 4c and 4c). We additionaly investigate the batch size in Figure 4d and 4h, a factor overlooked in previous work (Hamamci et al., 2024; Shui et al., 2025).

**CLS token.** When propagating the *cls token* across different attention levels, in addition to the default averaging setting, we also consider two alternative scenarios: (i) global average pooling at the end of the visual encoder (*none*); and (ii) applying weighted averaging using an attention pooling (*learned*). As shown in Figures 4a and 4e, aggregating the *cls token* by averaging proves to be the most effective approach. This result demonstrates that more complex components are not necessary to achieve better performance.

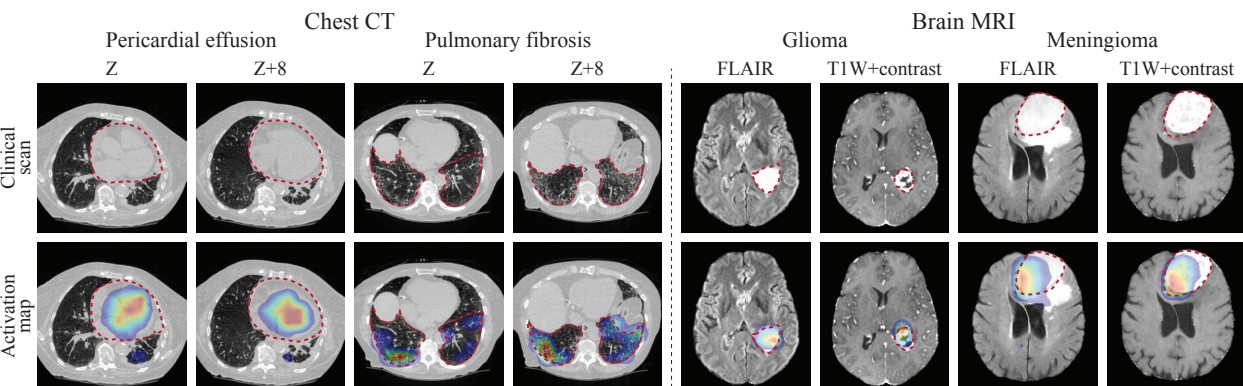

Figure 5: Qualitative results of zero-shot diagnosis on the Rad-ChestCT (Draelos et al., 2021) and BraTS (Baid et al., 2021; LaBella et al., 2023) datasets. The first row shows the original clinical scans with pathologic regions outlined (dashed red). The second row shows activation maps (Chefer et al., 2021). HLIP identifies the pathologic regions across multiple groups of adjacent chest CT slides (left) and brain MRI scans (right).

**Location of dense attention layers.** In Figure 4b and 4f, we consider alternative locations of the global attention layer (*i.e.*, study attention for uncurated datasets and scan attention for curated datasets). Compared with distributions such as placing all global attention layers at the front or all at the end, we find that evenly distributed global attention layers yield the best performance. This is because they enable the model to capture global information at multiple feature levels.

**Number of dense attention layers.** In Figure 4c and 4c, we investigate the number of global attention layers, where 0 and 12 correspond to two implementations of vanilla ViT. We find that 4 global attention layers are sufficient to capture global information, outperforming both 0 layers, which lack global information, and 12 layers, which lack the constructive priors provided by slice or scan attention.

**Batch size.** Previous work (Hamamci et al., 2024; Shui et al., 2025) used only small batch sizes (*e.g.*, 48) during the pre-training. In Figure 4d and 4h, we investigate this factor, which has been widely recognized as important in language-image pre-training, in the context of 3D medical imaging. We find that language-image pre-training in 3D medical imaging still benefits from larger batch sizes, underscoring the importance of memory-friendly designs such as our hierarchical attention mechanism.

### 4.5    Visualizations

In Figure 5, we visualize the activation maps of HLIP, following Chefer et al. (2021) in the zero-shot setting on Rad-ChestCT (Draelos et al., 2021) and BraTS (Baid et al., 2021; LaBella et al., 2023). Although HLIP mainly employs lightweight local attention layers, these visualization results demonstrate the effectiveness of a few global attention layers in enabling HLIP to attend to visual features globally. Specifically, for chest CT (curated, single scan), HLIP attends to pathologic regions across different slices, e.g., slice Z and Z+8, by leveraging scan attention. For brain MRI (uncurated, multiple scans), HLIP attends to pathologic regions across different scan types, e.g., FLAIR and T1W+contrast, by leveraging study attention. This demonstrates the effectiveness HLIP in modeling the hierarchical structure of 3D medical imaging.

### 4.6    Clinical Translation of HLIP

We conduct a 1-year, health system-scale, comprehensive evaluation on a prospective set of neurological studies. The datasets include ~23K brain MRI studies covering 52 diagnoses and ~15K head CT studies covering 83 diagnoses. We report macro AUC in Figure 6a and 6b. HLIP consistently outperforms ViT. Results for each diagnosis are presented in Figures 11 and 12 in Appendix C.

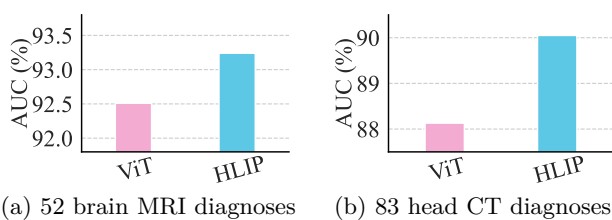

(a) 52 brain MRI diagnoses      (b) 83 head CT diagnoses

Figure 6: Prospective testing results on (a) 52 brain MRI diagnoses and (b) 83 head CT diagnoses.

## 5 Conclusion

In this work, we pioneer language-image pre-training directly on uncurated 3D medical imaging studies containing multiple scans, together with a novel hierarchical attention mechanism that effectively extracts visual features. These two contributions form HLIP, a scalable and effective pre-training framework for 3D medical imaging. Despite its simplicity in both concept and practice, HLIP achieves state-of-the-art performance on multiple benchmarks across diverse modalities and anatomical regions, including brain MRI, head CT, and chest CT. These results demonstrate that directly pre-training on uncurated clinical datasets is a scalable and effective paradigm for language-image pre-training in 3D medical imaging. We hope our work facilitates large-scale language-image pre-training in other health systems and inspires future machine learning research on scalable approaches for real-world clinical data.

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

## Appendix

## A  Dataset

As mentioned in Section 3.2, this section provides additional details on the two datasets we collected: BrainMRI220K and HeadCT240K. For other datasets, such as CT-RATE (Hamamci et al., 2024), Rad-ChestCT (Draelos et al., 2021), BraTS 2023 (Menze et al., 2014; Bakas et al., 2017; Baid et al., 2021; LaBella et al., 2023; Moawad et al., 2024; Kazerooni et al., 2024), NYU-Mets (Link et al., 2024), UCSF-Mets (Rudie et al., 2024), RSNA (Flanders et al., 2020), and CQ500 (Chilamkurthy et al., 2018), we refer readers to their original publications. Additionally, we include a discussion of our preprocessing strategy.

### A.1  BrainMRI220K

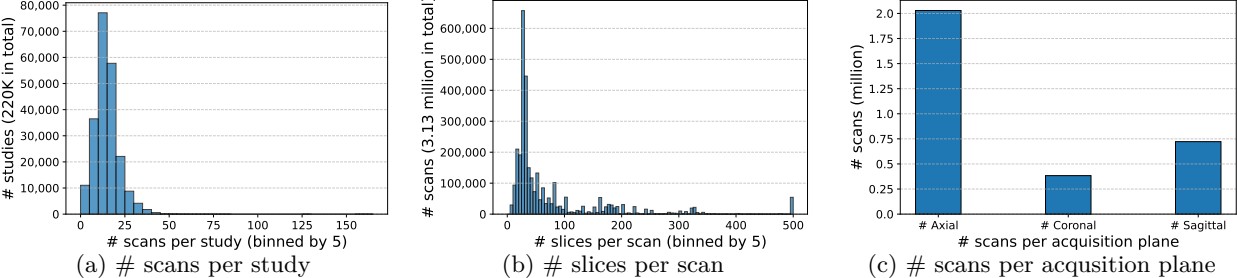

(a) # scans per study  (b) # slices per scan  (c) # scans per acqusition plane

Figure 7: Statistic results on BrainMRI220K.

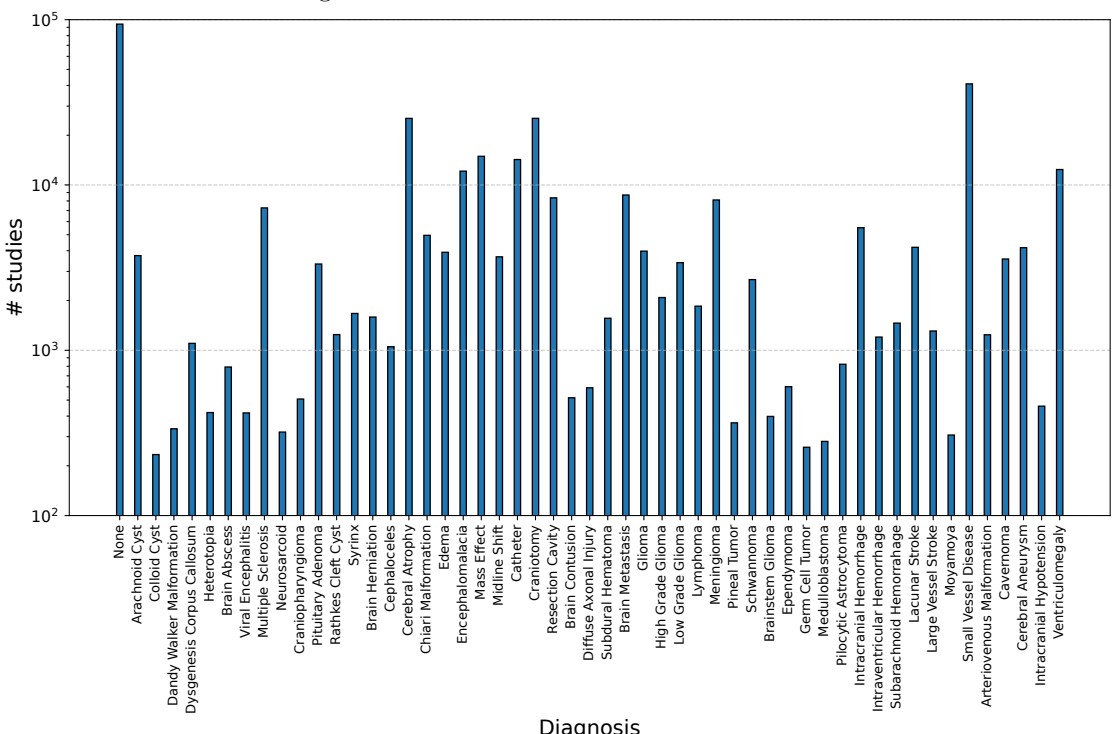

Figure 8: Diagnosis distribution on BrainMRI220K.

The distributions of the number of scans per study, slices per scan, and scans per acquisition view in the BrainMRI220K dataset are shown in Figure 7. Specifically, the number of scans per study ranges from 1 to 162, with the third quartile at 17. In total, there are 3.13 million scans from 220,001 MRI studies. All scans are acquired with through-plane spacing equal to or smaller than 4mm. The number of slices per scan ranges from 5 to 500, with the third quartile at 80. Each scan is originally acquired from a different view, with the ratio between axial, coronal, and sagittal views being 5:1:2. Moreover, we show the distribution of 52 brain MRI diagnoses in Figure 8, based on keyword statistics extracted from the reports.

## A.2 HeadCT240K

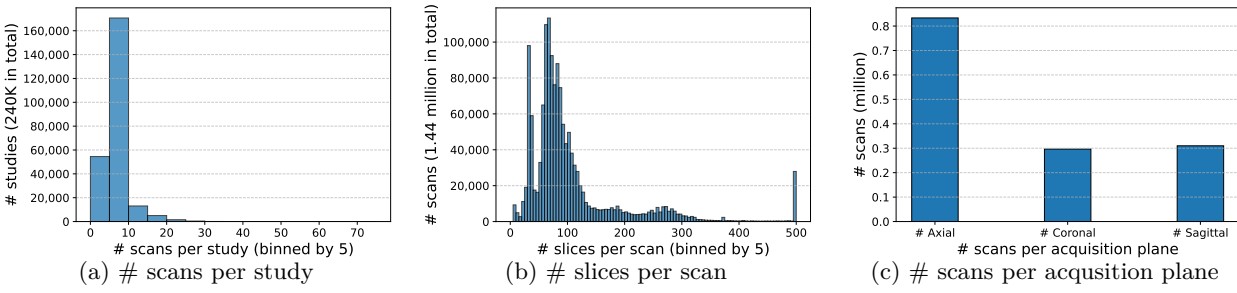

(a) # scans per study     (b) # slices per scan     (c) # scans per acqusition plane

Figure 9: Statistic results on HeadCT240K.

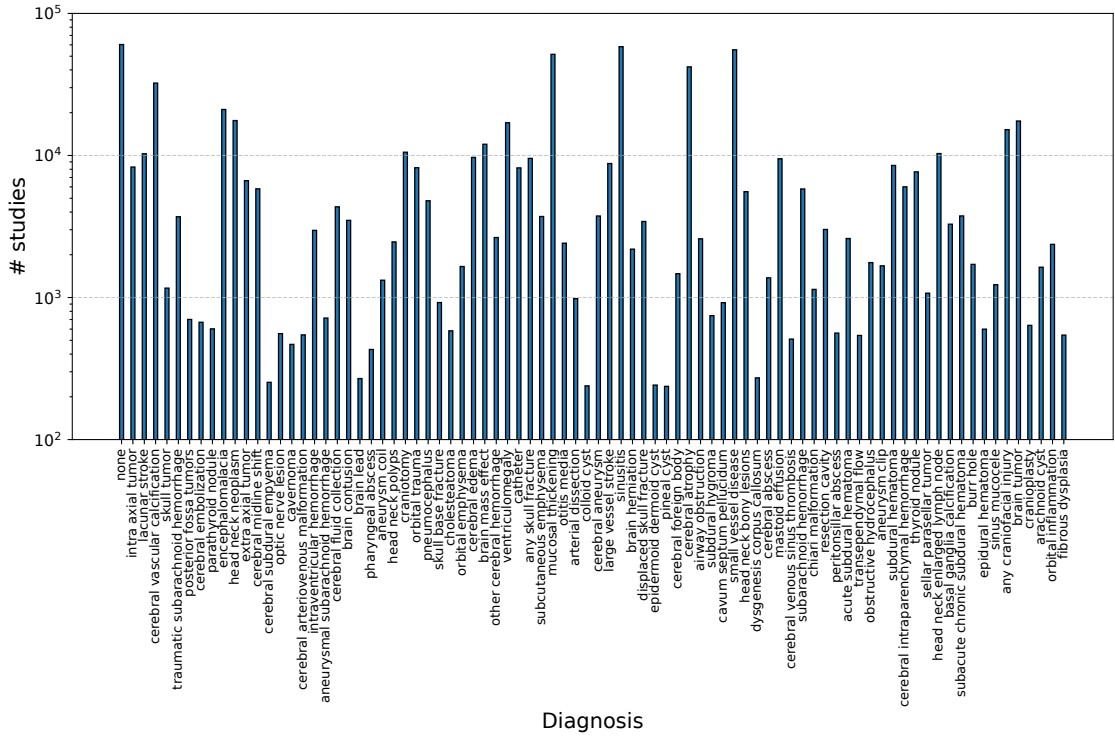

Figure 10: Diagnosis distribution on HeadCT240K.

The distributions of the number of scans per study, slices per scan, and scans per acquisition view in the HeadCT240K dataset are shown in Figure 9. Specifically, the number of scans per study ranges from 1 to 71, with the third quartile at 6. In total, there are 1.44 million scans from 244,253 CT studies. All scans are acquired with through-plane spacing equal to or smaller than 4mm. The number of slices per scan ranges from 5 to 500, with the third quartile at 110. Each scan is originally acquired from a different view, with the ratio between axial, coronal, and sagittal views being 8:3:3. Moreover, we show the distribution of 83 head CT diagnoses in Figure 10, based on keyword statistics extracted from the reports.

### A.3 Discussion

To the best of our knowledge, this is the first work to handle 3D medical datasets at the scale of Brain-MRI220K and HeadCT240K, which comprise millions of scans. Our preprocessing strategy, introduced in Section 3.2, follows standard practices where possible, but also diverges due to the scale and diversity of our dataset. Specifically, percentile clipping for brain MRI and HU value truncation for head CT follow standard practices (Ma et al., 2024; Zhu et al., 2025). For head CT, we do not concatenate scans with different HU truncations along the channel dimension, as our method is capable of processing multiple scans simultaneously. However, our strategy regarding orientation and spacing may differ from established preprocessing pipelines (Isensee et al., 2021; Blankemeier et al., 2024; Hamamci et al., 2024; Shui et al., 2025).

**Orientation.** CT and MRI scans are acquired using different planes, including axial, coronal, and sagittal. Isotropic data are rare in routine radiology studies; the through-plane spacing is typically larger than the in-plane spacing. In the context of batch construction, where all samples in a batch must share the same shape, standardizing the orientation can lead to downsampling along the in-plane axes and upsampling along the through-plane axis. Upsampling in pixel space does not introduce new information, whereas downsampling results in information loss. From the perspective of data augmentation, the orientation can change when 3D random rotation is applied (Isensee et al., 2021). So, *why do we standardize the orientation at the beginning then augment it during training*? Instead, we consistently align the first dimension (depth) with the through-plane axis. Please note that the in-plane orientation is still standardized as `RP`, `RI`, or `PI`. We treat the diversity of acquisition planes (as shown in Figure 7c and Figure 9c) in our dataset as a form of natural data augmentation.

**Spacing.** Standardizing spacing across the entire dataset is a common practice in medical image segmentation to ensure that convolution filters interpret anatomical structures consistently (Isensee et al., 2021). However, HyperSpace (Joutard et al., 2024) demonstrates that, with appropriate spacing augmentation, model can learn spacing-invariant features with sufficient spacing augmentation. Given the diversity of spacing settings (as shown in Figure 7b and Figure 9b) in our dataset, we believe it naturally occupies sufficient spacing augmentation for the model to learn spacing-invariant features. Therefore, when constructing a batch, we first resize each volume to `(48,256,256)`, and then apply a center crop to obtain `(48,224,224)`. Here, 48 represents the median number of slices, which does not substantially distort the original scan.

**M scans per study.** As the number of scans per study varies, we randomly sample $M$=10 scans per study at each training step to construct a batch. An equal number of positional embeddings is also sampled for these $M$ scans from a total of $M_{\max}$=40 positional embeddings at each step. This means the model is exposed to $M_{\max}$ different positions over the course of training, while observing $M$ positions at each step. This is analogous to a dropout strategy, with a scan dropout rate of $1 - \frac{M}{M_{\max}}$. During evaluation, the model is able to process studies containing up to $M_{\max}$ scans without the need for interpolation.

## B Implementation

In this section, we summarize the model configuration described in Section 3.2 and provide details of the pre-training setup referenced in both Section 3.2 and Section 4.3. For linear probing evaluation on head CT, we refer readers to FM-HeadCT (Zhu et al., 2025).

### B.1 Chest CT

The model architecture and pre-training configuration for chest CT are summarized in Table 3 and Table 4, respectively. These configurations closely follows prior works such as CT-CLIP (Hamamci et al., 2024), OpenCLIP (Cherti et al., 2023), and BiomedCLIP (Zhang et al., 2023a). In the pre-training configuration, we report the base learning rate corresponding to a batch size of 64. During training, we follow the linear learning rate scaling rule (Li et al., 2023): $lr = base\_lr \times \frac{batch\_size}{64}$.

The diseases used for zero-shot evaluation are listed in Table 5. This setup is consistent with fVLM (Shui et al., 2025) and all methods reported in Table 2.

Table 3: Model (Chest CT)

| config | value |
|---|---|
| projection | linear |
| embed dim | 512 |
| *Visual Encoder* | ViT-B (Dosovitskiy et al., 2020) |
| slice attn index | (0,1;3,4;6,7;9,10) |
| scan attn index | (2,5,8,11) |
| input size | (112,336,336) |
| patch size | (8,24,24) |
| patch dropout | 0.0 |
| *Text Encoder* | CXR-BERT (Boecking et al., 2022) |
| context length | 512 (Hamamci et al., 2024) |

Table 4: Pre-training (Chest CT)

| config | value |
|---|---|
| image precision | float32 |
| window | [-1150,350] |
| mean, std | [0.449,0.226] |
| report | original |
| optimizer | AdamW (Loshchilov & Hutter, 2017) |
| $\beta_1$, $\beta_2$ | 0.9, 0.98 (Radford et al., 2021) |
| base *lr* | 1e-5 |
| weight decay | 0.2 |
| *lr* schedule | cosine (Loshchilov & Hutter, 2016) |
| warmup steps | 47 (Shui et al., 2025) |
| numerical precision | amp |

Table 5: Evaluation (Chest CT)

| CT-RATE (Hamamci et al., 2024) | Rad-ChestCT (Draelos et al., 2021) |
|---|---|
| Emphysema | emphysema |
| Atelectasis | atelectasis |
| Lung nodule | nodule |
| Lung opacity | opacity |
| Pulmonary fibrotic sequela | fibrosis |
| Pleural effusion | pleural_effusion |
| Mosaic attenuation pattern | $\emptyset$ |
| Peribronchial thickening | bronchial_wall_thickening |
| Consolidation | consolidation |
| Bronchiectasis | bronchiectasis |
| Interlobular septal thickening | septal_thickening |
| Cardiomegaly | cardiomegaly |
| Pericardial effusion | pericardial_effusion |
| Coronary artery wall calcification | calcification |
| Hiatal hernia | hernia |
| Arterial wall calcification | calcification |

## B.2 Brain MRI/Head CT

Table 6: Model (Brain MRI/Head CT)

| config | value |
|---|---|
| projection | linear |
| embed dim | 512 |
| *Visual Encoder* | ViT-B (Dosovitskiy et al., 2020) |
| scan attn index | (0,1;3,4;6,7;9,10) |
| study attn index | (2,5,8,11) |
| input size | (48,224,224) |
| patch size | (8,16,16) |
| patch dropout | 0.25 |
| *Text Encoder* | PubMedBERT (Gu et al., 2020) |
| context length | 256 (Zhang et al., 2023a) |

Table 7: Pre-training (Brain MRI/Head CT)

| config | value |
|---|---|
| image precision | uint8 |
| truncate | [0.05,0.95] |
| mean, std | [0.449,0.226] |
| report | Prima (Lyu et al., 2026) |
| optimizer | AdamW (Loshchilov & Hutter, 2017) |
| $\beta_1$, $\beta_2$ | 0.9, 0.95 (Li et al., 2023) |
| base *lr* | 1e-4 |
| weight decay | 0.2 |
| *lr* schedule | cosine (Loshchilov & Hutter, 2016) |
| warmup steps | 2000 |
| numerical precision | amp |

The model architecture and pre-training configuration for brain MRI/head CT are summarized in Table 6 and Table 7, respectively. We process the reports using the method proposed in Prima (Lyu et al., 2026), which leverages a large language model to summarize the original radiologist reports and reduce biases related to writing style, grammar, and wording habits. Given the rapid development of modern language models, we do not consider this fully automatic text processing applied to existing radiologist reports to be equivalent to the data curation pipeline on the visual side, which relies on radiologists and imposes additional burdens.

## C Prospective Evaluation

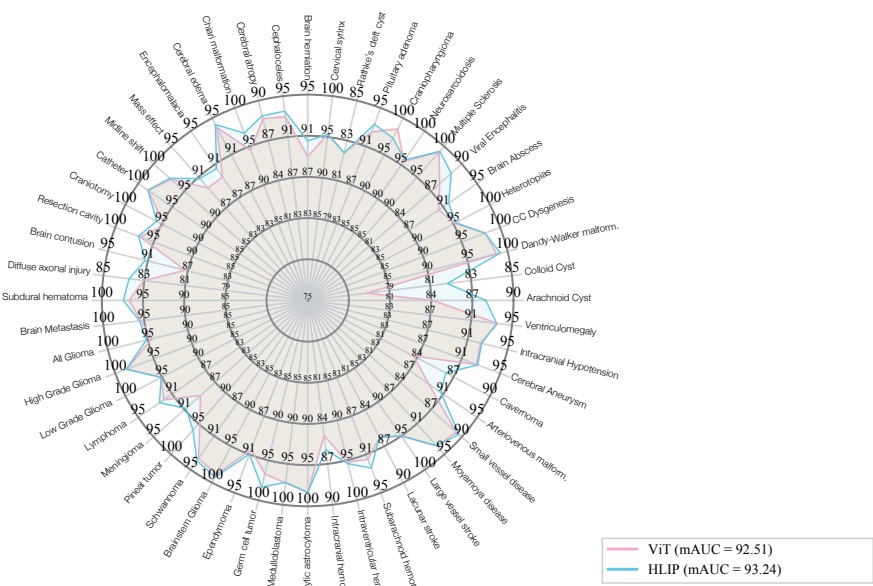

Figure 11: Prospective evaluation on 52 brain MRI diagnoses.

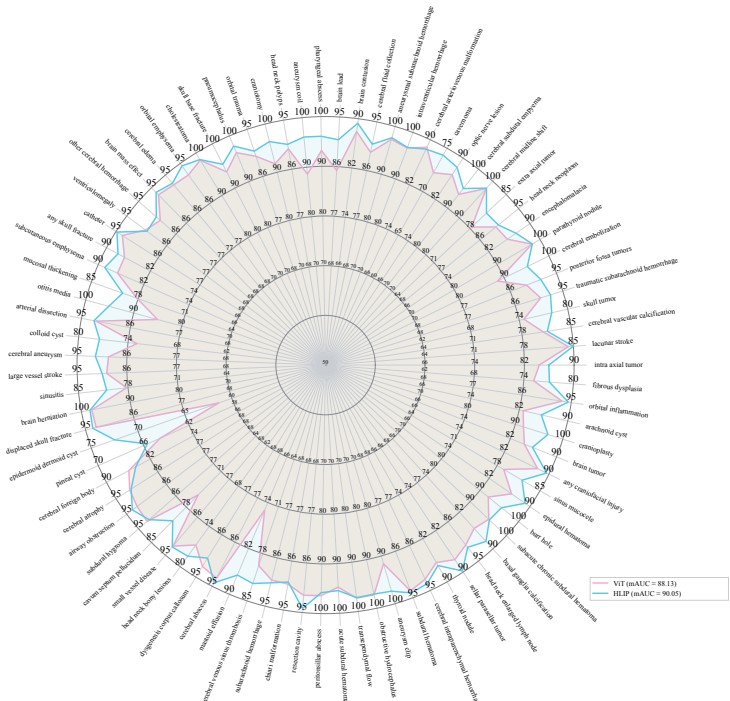

Figure 12: Prospective evaluation on 83 head CT diagnoses.

We present the detailed results of the prospective evaluation conducted within our health system for 52 brain MRI diagnoses and 83 head CT diagnoses in Figure 11 and Figure 12, respectively. We report linear-probe AUC for HLIP (blue) and the vanilla ViT (red). HLIP demonstrate consistent improvement.

## D    Statistical Analysis

We investigate statistical metrics for HLIP and the original ViT on the Pub-Brain-5 dataset. First, using the zero-shot prompt reported in the paper, "*This brain MRI shows: {disease}.*", we find that the p-value between the two models is less than 0.0001. We then perform inference using four additional prompts: "*This brain MRI shows: likely {disease}.*", "*This brain MRI shows: {disease} identified.*", "*This brain MRI shows: suggesting {disease}.*", and "*This brain MRI shows: compatible with {disease}.*" Under the five prompts considered, the 5-way disease classification performance is $63.4_{\pm 3.8}$ for HLIP and $40.6_{\pm 4.0}$ for ViT. This result demonstrates a clear benefit of the HLIP methodology over the original ViT.

Table 8: Results of zero-shot classification on Pub-Brain-5. We report the balanced accuracy of HLIP under five prompts.

| | Anomaly Detection | | | | | Tumor | Disease |
|---|---|---|---|---|---|---|---|
| | Stroke | Glioma | Meningioma | Metastasis | *mean* | | |
| Pub-Brain-5 | $86.9 \pm 6.0$ | $92.4 \pm 4.8$ | $76.1 \pm 3.4$ | $84.9 \pm 4.9$ | $85.4 \pm 3.7$ | $62.3 \pm 1.4$ | $63.4 \pm 3.8$ |
| Pub-Brain-5-GT | $90.4 \pm 5.2$ | $92.5 \pm 5.0$ | $76.3 \pm 3.3$ | $77.4 \pm 6.9$ | $84.1 \pm 3.7$ | $53.8 \pm 0.9$ | $59.3 \pm 3.7$ |

In Table 8, we additionally report the full results of HLIP under this setting on the Pub-Brain-5 and Pub-Brain-5-GT benchmark, for reference in future work.

## E    Complexity

In this section, we analyze the computational efficiency of the proposed hierarchical attention mechanism from both theoretical and practical perspectives.

### E.1    Analysis

Given a sequence length of $N$, model dimension of $c$, the matrices $Q, K, V \in \mathbb{R}^{N \times c}$. The output $O \in \mathbb{R}^{N \times c}$ of standard self-attention is computed as:

$$S = Q \times K^T \in \mathbb{R}^{N \times N}; \quad P = softmax(S) \in \mathbb{R}^{N \times N}; \quad O = PV \tag{1}$$

where $S$ and $P$ require $\Omega(N^2)$ memory access; $Q, K, V, O$ requires $\Omega(N \times c)$ memory access. Therefore, the I/O complexity for a standard self-attention over $N$ tokens is:

$$\Omega(N^2 + N \times c) \tag{2}$$

The $S$ requires $\Omega(N^2 \times c)$ multiply-add operations; $P$ requires $\Omega(N^2)$ multiply-add operations; $O$ requires $\Omega(N^2 \times c)$ multiply-add operations. Therefore the compute complexity for a standard self-attention over $N$ tokens is:

$$\Omega(N^2 \times c) \tag{3}$$

**Study Attention.** In our scenario, the sequence length of a study is $N = M \times d \times h \times w$. The study attention is a standard self-attention over all $N$ tokens. Therefore the I/O complexity and the compute complexity of the study attention is $\Omega(N^2 + N \times c)$ and $\Omega(N^2 \times c)$, respectively, as shown in Equation 2 and Equation 3.

**Scan Attention.** The scan attention is $M$ standard self-attention operations, each over $\frac{N}{M}$ tokens. Therefore replace $N$ with $\frac{N}{M}$ in Equation 2 and Equation 3 resulting in $\Omega(\frac{N^2}{M^2} + \frac{N \times c}{M})$ and $\Omega(\frac{N^2}{M^2} \times c)$ for I/O and compute complexity of each self-attention operations, therefore the total I/O and compute complexity of the scan attention are $\Omega(\frac{N^2}{M} + N \times c)$ and $\Omega(\frac{N^2}{M} \times c)$, respectively.

**Slice Attention.** The slice attention is $M \times d$ standard self-attention operations, each over $\frac{N}{M \times d}$ tokens. Therefore, similar to the scan attention the total I/O and compute complexity of the slice attention are $\Omega(\frac{N^2}{M \times d} + N \times c)$ and $\Omega(\frac{N^2}{M \times d} \times c)$, respectively.

### E.2 Experiments

We conduct experiments comparing ViT (Dosovitskiy et al., 2020), the 3D Swin Transformer implemented in MONAI (Cardoso et al., 2022), and our HLIP visual encoder. Since Swin is not compatible with multi-scan inputs, the experiment is conducted on a single-channel 3D input of shape (B,1,224,224,224), where B=1 denotes the batch size. Both ViT and HLIP use a ViT-Base architecture with patch_size=(16,16,16). The Swin Transformer is configured with dim=128, patch_size=(4,4,4), window_size=(7,7,7), depth=(2,2,12,2), and head=(4,8,16,32). Following the original Swin Transformer (Liu et al., 2021) while matching the model size.

Table 9: Practical Complexity.

|  | model size (M) | throughput (img/s; ↑=better) | memory (G; ↓=better) |
|---|---|---|---|
| ViT | 88.2 | 9.0 | 6.6 |
| Swin | 87.9 | 9.9 | 8.0 |
| HLIP (ours) | 88.2 | 17.5 | 4.1 |

All experiments are conducted on a single A40 GPU. We report the model size, throughput and training memory for each model. To measure throughput, we include 100 warm-up iterations and compute the average over 1000 forward passes. As shown in Table 9, Swin is slightly faster than the original ViT but lags behind our HLIP visual encoder. Swin also consumes more memory during training due to the large codebook of relative position embeddings constructed for 3D inputs. Moreover, Swin is not compatible with flash attention, which has been integrated into PyTorch 2.0 and supports a wide range of GPUs.

## F Discussion

In this work, we present HLIP, a scalable language–image pre-training framework for 3D medical imaging. For health systems that have accumulated large volumes of data over the past decades, HLIP offers a new direction for learning transferable representations. In this section, we provide additional discussion, including both the limitations and future directions for readers interested in pursuing this line of research.

**Study Curation.** In this work, we collect all studies from our health system, resulting in an imbalanced pre-training dataset. Building on findings from the natural image domain, we believe that developing a systematic approach to construct a more balanced pre-training dataset is an important direction for future work. Please note that this curation should occur at the study level and therefore does not impose additional burden on radiologists, unlike selecting representative scans or slices.

**Zero-shot Transferability.** We observe that zero-shot transferability does not always correlate with the number of training studies, in contrast to the findings of Udandarao et al. (2024). For example, although keyword search yields more patients with meningioma or metastasis than with glioma, zero-shot performance on glioma is substantially higher. We hypothesize that this discrepancy arises from fundamental differences between natural image and medical imaging datasets. For natural image datasets such as ImageNet, the relationship between the number of positive instances in the training corpus and zero-shot performance is approximately log-linear. This observation implicitly assumes a small or near-zero Bayes error rate, for example, when differentiating boats from dogs. In contrast, in medical imaging, diagnosing meningioma and metastasis is challenging for multifaceted reasons, including lesion size, radiographic features, and anatomical location, which leads to a higher error rate compared to glioma.

**Scale.** Even at its current scale of 200K studies, the largest to date, our dataset remains far smaller than training scales in the natural image domain, which typically involve billions of samples. Although collected from a neuro-radiology system, we find that other organ systems such as the spine, abdomen, and knee occasionally appear in our dataset, and the current model demonstrates promising zero-shot transferability to these organs as well. Therefore, we believe there is no barrier to training a model on more comprehensive data, including different organ systems (*e.g.*, brain, cardiac, chest, abdomen) and modalities (*e.g.*, CT and MRI). Since no human labeling burden is involved, we expect the training scale to expand to millions of samples in the near future.

As the dataset scales further, especially when encompassing multiple organ systems and modalities, the underlying patterns will become more complex. Therefore, we believe that larger batch sizes will be necessary. Although our hierarchical attention mechanism introduces no additional computational burden, we rely on both flash attention and gradient checkpointing to achieve a batch size of 256 on 8 L40 GPUs. Further increasing the batch size will require either a gradient accumulation strategy or additional computational resources.

Moreover, as the dataset scales up, larger models may prove beneficial but will also require additional computational resources.

**Vision-Language Model.** Although current vision–language models achieve impressive performance even on medical tasks, we believe these general-purpose models will reach a bottleneck when sufficient specialist medical data are not available on the internet. On the other hand, as with other language-supervised visual encoders, HLIP's visual encoder could facilitate the development of vision language models under current frameworks. Therefore, we expect to see a radiology specialist vision–language model that can operate on entire studies in the near future.

**Dataset Release Policy.** The assets introduced in this work, including a new brain MRI benchmark for zero-shot classification, an effective language–image pre-training implementation for 3D medical imaging, the pre-training recipe, and the model checkpoints, will be fully publicly available. However, we are unable to release the BrainMRI220K and HeadCT240K datasets alongside this work due to privacy constraints associated with neuroimaging data. We plan to release a synthetic dataset in the future, where the generative model is trained on our large-scale dataset.

