# OpenReview forum: "Towards Scalable Language-Image Pre-training for 3D Medical Imaging"
_TMLR — Accepted by TMLR_

### Review · Reviewer_TJLi · 2025-10-16

**Summary Of Contributions:**

### Summary:
The authors' contribution to the community is three-fold: a) they introduce a lightweight attention mechanism tailored to the inherent structure of the data, b) they present a scalable pre-training approach that does not rely on human-annotated, curated data, and c) they provide a zero-shot dataset for brain MRI disease classification. The authors evaluate their approach on both uncurated and curated data, including a zero-shot experiment and a real-world case study.

### Strengths:

**(S1 - clarity and quality wrt problem setting).** The introduction and related work sections are very well written. They clearly define the general problem of pretraining in 3D medical imaging and discuss the limitations of existing SOTA methods. This effectively sets the stage for the authors’ contributions, as the need for methods that can handle uncurated data becomes evident, and the motivation for a more lightweight attention mechanism is well justified given the large number of tokens and samples involved.

**(S2 - description of the input data format and the attention mechanism).** I highly appreciate the first paragraph of Section 3 and Section 3.1. The data object "study" is well introduced, with all dimensions clearly explained. The descriptions of "volume" elements and "visual tokens" make the input format for training and evaluation easy to understand. The explanation of the different dimensions further aids in grasping the data structure and how information is shared across the various attention mechanism variants.

**(S3 - relevance and novelty of the attention mechanism variants proposed).** The proposed lightweight attention mechanism is both novel and relevant. It is well motivated: replacing full attention with a more efficient version reduces computational overhead, and the included inductive bias is grounded in the data's structure (consider a) a whole data sample, b) scan-wise, or c) share information across tokens within slides close to each other). I also appreciate the discussion of similarities and differences with sliding-window attention.

**(S4 - strong results).** The proposed method outperforms baselines which may indicat its effectiveness. However, see (W2) and (Questions).

### Weaknesses:

**(W1 - clarity wrt to pre-training).** If understood correctly, a) the proposed model leverages uncurated (i.e., non-annotated) data, and b) pretraining is performed via contrastive learning by matching text and image representations. However, it is unclear what exactly serves as input to the text encoder. The manuscript lacks sufficient detail in this regard, which is critical since pretraining on uncurated data is a major selling point of the work.

**(W2 - missing error bars).** All results are reported without error bars or statistical tests. This significantly weakens the strength of the reported findings, as the observed differences may be due to random variation. The authors should include error bars and appropriate significance testing.

**(W3 - selection of attention-mechanism hierarchy).** The authors describe mixing blocks with different attention hierarchies, which is reasonable and reminiscent of mixed-architecture LLMs using SSM and transformer blocks. However, it would be helpful to provide intuition on when to choose scan-wise vs. adjacent-slice-wise attention. How was this decision made? Was it treated purely as a hyperparameter? More insight would be valuable.

### Questions:
* See W1 and W3.
* The model is pretrained on two in-house datasets, BrainMRI220K and HeadCT240K. Will these datasets be released? If not, reproducibility is limited, and the authors should consider using public datasets.

### Very minor comment:
max, mae, scan, slice, and any other abbreviations or terms appearing in formulas should be written in non-italic font.

**Audience:**

Yes

**Audience Explanation:**

Medical imaging and machine learning methods tailored to it are of interest to the community. The authors clearly articulate the limitations of current state-of-the-art approaches, and the proposed lightweight attention mechanism appears both reasonable and interesting.

**Broader Impact Concerns:**

No concerns.

**Claims And Evidence:**

No

**Claims Explanation:**

Unfortunately, the manuscript lacks details on how the preprocessing was performed, making it difficult to assess how the central research question — training from uncurated data — was addressed. In addition, the effectiveness of the proposed approach cannot be verified due to the absence of statistical tests and error bars.

**Requested Changes:**

- Clarify the pre-training procedure, including data inputs and methodology.
- Add error bars and statistical tests to all reported results in tables and figures.
- If not already planned, open-source the datasets used for pre-training or consider using publicly available alternatives.

---

> ### Author Response · Authors · 2026-01-01
> **Response to reviewer TJLi (Part 1)**
>
> Thank you for your constructive feedback. Here is our reply to your questions.
>
> ---
>
> ### W1: The input to the text encoder.
>
> The input to the text encoder is the radiologist's report that is paired with the radiology images prior to any data curation.
>
> Without any text pre-processing, supervision using the raw radiology reports, where only the Findings and Impression sections are included, achieves performance comparable to that obtained using large language model summarized reports, as shown in Table 2.
>
> In the BrainMRI220K and HeadCT240K datasets, we process the reports using the method proposed in Prima [1], which leverages a large language model to summarize the original radiologist reports and reduce biases related to writing style, grammar, and wording habits. Given the rapid development of modern language models, we do not consider this fully automatic text processing applied to existing radiologist reports to be equivalent to the data curation pipeline on the visual side, which relies on radiologists and imposes additional burdens. We have highlighted this clarification in Appendix B.2 in the revised manuscript.
>
> ---
>
> ### W2: Error bars
>
> We investigate statistical metrics for HLIP and the original ViT on the Pub-Brain-5 dataset. First, using the zero-shot prompt reported in the paper, “This brain MRI shows: {disease}.”, we find that the p-value between the two models is **less than 0.0001**. We then perform inference using four additional prompts: “This brain MRI shows: likely {diseases}.”, “This brain MRI shows: {diseases} identified.”, “This brain MRI shows: suggesting {diseases}.”, and “This brain MRI shows: compatible with {diseases}.” Under these five prompts, the 5-way disease classification performance is **63.4 (+3.8, −3.8)** for HLIP and **40.6 (+4.0, −4.0)** for ViT. This result demonstrates a clear benefit of the HLIP methodology over the original ViT. However, we do not include error bars in the main experiments for the following reasons:
>
> - First, the performance gaps between our method and the baseline methods are substantial. For example, compared with the original ViT trained under the same setting, we observe a +20.5% accuracy improvement on the Pub-Brain-5 dataset for brain MRI and a +2.4% AUC improvement on the RSNA dataset for head CT. HLIP further demonstrates a +3.7% AUC improvement on the Rad-ChestCT dataset for chest CT when compared with current state-of-the-art methods. These evaluation datasets also contain thousands of cases, including approximately 18K samples in Pub-Brain-5, 10K samples in RSNA, and 3K samples in Rad-ChestCT. Given the magnitude of these improvements and the scale of the evaluation datasets, we believe that the observed gains are unlikely to arise from random factors unrelated to our method.
>
> - Second, the majority of our experiments focus on zero-shot classification. Unlike evaluation protocols that require additional fine-tuning, zero-shot evaluation does not introduce extra bias from task-specific training, which further supports the stability of the reported results.
>
> - Finally, during evaluation, we strictly follow the protocols specified by the baseline methods, including fVLM, CT-CLIP, BiomedCLIP, and ConceptCLIP. None of these baselines reports error bars, and we adhere to the same evaluation practice for consistency.
>
> ---
>
> ### W3: Selection of attention-mechanism hierarchy
>
> The choice of slice-, scan-, or study-level attention is treated as a hyperparameter, with selection guided by both intuitive considerations and systematic hyperparameter search.
>
> - From an intuitive perspective, uncurated radiology studies that contain multiple scans, such as T1, T2, and FLAIR, as well as different views, including axial, sagittal, and coronal, are likely to benefit from study-level attention rather than scan-level attention alone, as study attention facilitates comparison of radiographic features across scans. In contrast, curated radiology studies that contain a single-volume medical image, such as chest CT from public datasets, are more likely to benefit from scan-level attention rather than slice-level attention alone, as it enables effective aggregation of information across slices.
> - At the same time, Figures 4c and 4g show that introducing too many global attention layers can be detrimental, which further highlights the importance of careful hyperparameter selection.
>
> Regarding the placement of global attention layers, prior work, such as feature pyramid networks, aggregates features from multiple representation levels. Similarly, we hypothesize that HLIP benefits from aggregating global features across different layers. Figures 4b and 4f support this hypothesis for both uncurated and curated radiology datasets.
>
> ---
>
> **references**
>
> [1] Todd Hollon, et al. Learning neuroimaging models from health system-scale data. Research Square, pp. rs–3, 2025.

---

> ### Author Response · Authors · 2026-01-01
> **Response to reviewer TJLi (Part 2)**
>
> ### Q1: Plans for releasing the dataset.
>
> Currently, we are unable to release the BrainMRI220K and HeadCT240K datasets due to privacy constraints associated with neuroimaging data. However, the brain MRI benchmark for zero-shot classification, an effective language–image pre-training implementation for 3D medical imaging, the pre-training recipe, and the model checkpoints will be fully publicly available. These resources can still benefit researchers who have access to large-scale 3D medical datasets or who wish to leverage the released large-scale pre-trained models.
>
> Moreover, the datasets, CT-RATE and Rad-ChestCT, used in Table 2 are fully publicly available, ensuring reproducibility.

---

> ### Comment · Reviewer_TJLi · 2026-01-08
> **Response to authors**
>
> Dear authors,
> thank you for your responses.
>
> Please find my comments and questions below.
>
> **W1**: Thank you for the clarification, which helped me better understand your experimental setup.
>
> **W2**:
> - It is great to see p-values and performance intervals reported on the Pub-Brain-5 dataset.
> - I share your belief that the observed performance gains are very unlikely to stem from random factors. I also agree with both points you raised: a) for zero-shot experiments, there is no model re-run dimension over which performance variability needs to be tracked, and b) given ~18k samples, the standard deviation across samples should be small relative to the reported gains. Still, I do not see a strong reason why including error bars would not be beneficial:
>     * Even if it is unlikely that the gains stem from random factors, omitting error bars leaves room for unnecessary speculation. Including them would strengthen your already strong results and make them more convincing.
>     * Even when performance gains are large, best research practice suggests reporting statistical uncertainty where possible.
>     * I understand that you follow the evaluation protocol of prior work, where error bars were not included. From a practical perspective, I also understand that tables in the main manuscript are often easier to present without them. However, since your model sets a new state of the art and future work will likely compare against it, reporting error bars would be particularly valuable. If including them in the main manuscript is not feasible, would you consider adding them to the appendix?
>
> **W3**: Thank you for your thoughts and clarifications here.
>
> **Dataset release**: If I understand this correctly, the pre-training data will not be released, meaning the model cannot be fully reproduced. If so, this would significantly reduce the relevance of the work for the community. Are these limitations somewhere stated in the manuscript?

---

> ### Author Response · Authors · 2026-01-09
> **Response to reviewer TJLi (Part 3)**
>
> Thank you for agreeing with some of our arguments. Here is our response to unaddressed questions.
>
> ---
>
> ### W2: Statistical Analysis
>
> We agree with the reviewer that including statistical analysis is important when establishing a new benchmark. Therefore, we have included this statistical analysis, along with the full results of HLIP under the five prompts considered, in the Appendix (current D).
>
> ---
>
> ### Q1: Dataset release
>
> We appreciate the reviewer’s point regarding dataset release. The datasets were collected under IRB approval and are governed by institutional data use agreements; however, these agreements do not permit public release. In addition, neuroimaging data carry inherent re-identification risks, which further preclude unrestricted dissemination. As a result, releasing medical imaging data is particularly challenging, especially for neuroimaging. Looking ahead, we plan to release a synthetic dataset generated by a model trained on our data, which we hope will provide a practical way for the community to benefit from large-scale datasets of this kind. We have added this limitation to the discussion section in the Appendix (current F). However, we believe that HLIP demonstrates substantial impact even without releasing the datasets alongside this work:
>
> - **Generalizability of the proposed methodology.** Language–image pre-training has been highly successful in the natural image domain, largely because it can effectively leverage uncurated data at scale. HLIP is the first work to extend this paradigm to 3D medical imaging by exploring a scalable pre-training framework together with more effective modeling strategies. Importantly, the proposed methodology is not limited to BrainMRI220K or HeadCT240K and can be applied to other large-scale 3D medical imaging datasets. This provides a promising direction for researchers who have access to such data but have not been able to fully exploit it due to the burden of manual data curation.
>
> - **Value of the released model checkpoints.** The released checkpoints are also valuable. To the best of our knowledge, there is currently no publicly available 3D medical imaging model trained at a comparable scale. Making these checkpoints public can therefore benefit researchers seeking a strong visual encoder for a wide range of downstream tasks.
>
> - **Practical and efficient implementation.** Finally, our effective implementation of language-image pre-training for 3D medical imaging is itself a valuable asset to the community. On the public CT-RATE dataset, pre-training with our implementation takes approximately 6 hours using 4 A40 GPUs, which is substantially more efficient than existing implementations such as fVLM and CT-CLIP, which rely on 4 A100 GPUs and require significantly longer training time. We believe this efficiency can benefit researchers with limited computational resources and help accelerate progress in language–image pre-training for 3D medical imaging.

---

> > ### Comment · Reviewer_TJLi · 2026-01-11
> > **Reviewer's final comments**
> >
> > Comments on rebuttal:
> > * **W2: Statistical Analysis**. Thank you for adding error bars for the Pub-Brain-5 experiment. An analogous analysis for Pub-Brain-5-GT and CT-RATE (Table 2) is still missing. Since (a) I have already outlined my perspective that all experiments should ideally be supported by error bars, and (b) error bars are currently included for only one sub-experiment, I anticipate that the authors may disagree with me on their necessity. I respect this position and therefore only reiterate that I would have appreciated seeing all experiments supported by statistical analysis, and leave it to the other reviewers and the editor to decide on this matter.
> >
> > * **Q1: Dataset release**. Adding this limitation to the Appendix as well as the plan to release synthetic data is helpful. I would have preferred this limitation to be clearly stated in the main manuscript as well; nevertheless, I consider my concern to be (somewhat) addressed.
> >
> > **Summary**
> >
> > In summary, I consider this work interesting for the community and technically sound, with minor weaknesses in clarity, quality (missing error bars), and reproducibility. As the strengths outweigh the weaknesses, I lean towards acceptance (see recommendation).

---

### Review · Reviewer_jasq · 2025-11-03

**Summary Of Contributions:**

**Summary**

HLIP introduces a simple hierarchical attention schedule (slice, scan, study) that lets a ViT-style encoder pre-train directly on uncurated, multi-scan radiology studies. Trained at health-system scale (BrainMRI220K, HeadCT240K), it achieves SOTA zero-shot and linear-probe transfer performance on public benchmarks (Pub-Brain-5/5-GT, CQ500, RSNA, CT-RATE, Rad-ChestCT) and strong prospective results on large internal cohorts. The paper also contributes the Pub-Brain-5/5-GT evaluation sets, reports clear efficiency gains over ViT and Swin, and provides ablations on attention placement, class-token propagation, and batching.

**Strengths**

1. Clear problem setting and novelty: modeling uncurated, multi-scan studies and aligning the attention hierarchy with the radiology data hierarchy is reasonable and practically motivated.

2. Strong empirical results: the proposed model achieves consistent gains across diverse modalities (brain MRI, head CT, chest CT) in zero-shot and linear-probe settings, e.g., notably +10.5% balanced accuracy on Pub-Brain-5 disease classification and AUC gains on CQ500 and RSNA over strong head-CT baselines.

3. Scalability and efficiency: memory advantages over Swin and compatibility with flash attention and patch dropout are demonstrated. Ablations are informative.

4. Practical relevance: results on an internal prospective set, and qualitative activation maps that localize pathologies across slices and scans, strengthen translational value.

**Weaknesses**

1. Single-Source Data: The training data was collected from a single health system. It may introduce biases related to specific scanner manufacturers, imaging protocols, or patient demographics, which could limit the model's generalizability.

2. The authors observe that zero-shot transferability does not always correlate with the number of training studies for a given diagnosis. This discrepancy is not fully explained.

**Additional Comments:**

None.

**Audience:**

Yes

**Audience Explanation:**

3D Medical Imaging pretraining is a important topics for real-world applications.

**Broader Impact Concerns:**

None.

**Claims And Evidence:**

Yes

**Claims Explanation:**

1. The paper states concrete dataset sizes, model configs, and training recipes. It also includes ablations and shows consistent gains across Pub-Brain-5/5-GT, CQ500/RSNA, and CT-RATE/Rad-ChestCT

2. Convincing evidence: Improvements are large (e.g., 8.3% macro AUC on CQ500) and align with the proposed mechanism that hierarchical attention reduces attention cost while preserving global context intermittently.

**Requested Changes:**

See weakness 2.

---

> ### Author Response · Authors · 2026-01-01
> **Response to reviewer jasq**
>
> Thank you for your constructive feedback. Here is our reply to your questions.
>
> ---
>
> ### W1: Single-source data
>
> We appreciate this astute point raised by the reviewer.
>
> We agree that training on data from a single health system has the potential to limit generalization. However, we emphasize that BrainMRI220K and HeadCT240K include data from over 100 individual hospitals and imaging centers, thereby capturing all major scanner manufacturers (Siemens, Philips, GE), a wide range of imaging protocols, and diverse patient demographics. To provide evidence that HLIP is robust to this concern, we report external evaluations using data from other health systems in Table 1 and Figure 3 for brain MRI and head CT. We also present external evaluation results for chest CT in Table 2. Taken together, these results demonstrate that HLIP generalizes to other health systems and datasets and is robust to distribution shift.
>
> ---
>
> ### W2: Zero-shot transferability
>
> We hypothesize that this discrepancy arises from fundamental differences between natural image and medical imaging datasets. For natural image datasets such as ImageNet, the relationship between the number of positive instances in the training corpus and zero-shot performance is approximately log-linear [1]. This observation implicitly assumes a small or near-zero Bayes error rate, for example, when differentiating boats from dogs. In contrast, in medical imaging, diagnosing meningioma and metastasis is challenging for multifaceted reasons, including lesion size, radiographic features, and anatomical location, which leads to a higher error rate compared to glioma. We have highlighted this clarification in Appendix E of the revised manuscript.
>
> ---
>
> **references**
>
> [1] Vishaal Udandarao, et al. No" zero-shot" without exponential data: Pretraining concept frequency determines multimodal model performance. In The Thirty-eighth Annual Conference on Neural Information Processing Systems, 2024.

---

> ### Author Response · Authors · 2026-01-22
>
> We sincerely thank you for your great efforts in reviewing this paper. We have gone through your points one by one and tried to address them carefully. Please don’t hesitate to let us know if you have any further questions.

---

### Review · Reviewer_NGWG · 2025-12-25

**Summary Of Contributions:**

This paper proposes HLIP (Hierarchical attention for Language-Image Pre-training), a language–image pre-training approach tailored to real-world 3D radiology studies, such as brain MRIs and CT scans. The paper is motivated by the fact that prior 3D medical language–image pre-training often depends on curated data (single standardized scan per study, careful selection, extensive data cleaning), which is expensive and limits scale. HLIP instead aims to pre-train directly on uncurated clinical studies using a heirarchical attention mechanism that computes attention on three levels of granularity:
1. Slice-level attention: attention within groups of adjacent slices (lowest-level local context).
2. Scan-level attention: attention within each scan (mid-level context).
3. Study-level attention: occasional global attention across the whole study (cross-scan context).

The paper then performs training using two new uncurated datasets:
BrainMRI220K: ~221k MRI studies
HeadCT240K: ~244k CT studies

This paper also proposes a new benchmark, Pub-Brain-5, for zero-shot transfer evaluation.

Strengths
- I find the heirarchical attention mechanism to be a strong contribution, as it allows for a more nuanced approach to CLIP-style training. It allows for a larger batch size of training without the need to equally scale resources. It also demonstrates improved performance over contemporary methods, such as fVLM and CT-CLIP.
- Large scale data training strongly positions this work's contributions.

Weaknesses/questions
- When comparing performance of chest CT models, I am unclear if the training on CT-RATE takes place for a model checkpoint that is already trained on the HeadCT dataset or if training takes place from scratch from a randomly initialized model. If it is the case that a pretrained checkpoint is used, I am unclear the extent to which the model architecture itself is improving performance, and to what extent it is just a case of the model having seen more data, albeit of a different location.
- Was any preprocessing applied to the reports themselves in your two new datasets, or were raw reports used as is?
- In practice, DICOM series boundaries can be noisy, incomplete, or inconsistent, especially when working across institutions. Reporting styles may also differ. Were any efforts made to make training more robust to such cases? Do you think it would make a difference?

**Audience:**

Yes

**Audience Explanation:**

Yes I believe so. I believe the techniques, model, and benchmark introduced in this paper would definitely be useful for people working in this space, particularly for downstream tasks such as vision language modelling for medical images and segmentation tasks.

**Claims And Evidence:**

Yes

**Claims Explanation:**

Yes I believe they are
1. Large scale training is conducted using dataset sizes greater than existing methods.
2. Strong competing baselines are compared against in both head and chest settings.
3. Ablation studies confirm that the reported configuration of the model provides the strongest results.

**Requested Changes:**

Please provide clarity on the weakness and questions listed above.

---

> ### Author Response · Authors · 2026-01-01
> **Response to reviewer NGWG**
>
> Thank you for your constructive feedback. Here is our reply to your questions.
>
> ---
>
> ### W1: Model initialization for chest CT
>
> As described in Section 3.2, our model is initialized from an MAE pre-trained ViT-Base, which is the same initialization setting used by fVLM. This configuration is consistently applied across all three experiments, including brain MRI, head CT, and chest CT. No data or pre-trained weights are shared across these experiments. We apologize for the lack of clarity on this point. We have highlighted this clarification in Section 4.3 in the revised manuscript.
>
> ---
>
> ### W2: Report pre-processing
>
> We process the reports using the method proposed in Prima [1], which leverages a large language model to summarize the original radiologist reports and reduce biases related to writing style, grammar, and wording habits. Given the rapid development of modern language models, we do not consider this fully automatic text processing applied to existing radiologist reports to be equivalent to the data curation pipeline on the visual side, which relies on radiologists and imposes additional burdens. We apologize for the lack of clarity on this point. We have highlighted this clarification in Appendix B.2 in the revised manuscript.
>
> ---
>
> ### W3: More pre-processing details
>
> We appreciate this astute point raised by the reviewer.
>
> **DICOM**
>
> Medical imaging data, stored as DICOM files, can exhibit artifacts at image boundaries, including aliasing, wrap-around artifacts, and the presence of head devices or cushions. Moreover, these artifacts can vary across institutions, which is particularly relevant for our BrainMRI220K and HeadCT240K datasets that include MRI and CT scans collected from over 100 hospitals. Data at this scale presents both challenges and opportunities. Hand-engineered image processing methods for noise removal or data filtering often struggle in such unconstrained settings. At the same time, data collected in this setting captures the variability encountered in real-world clinical practice, including differences in scanner manufacturers, imaging protocols, and radiology reporting styles. Effectively modeling data at the scale of an entire health system can therefore help mitigate these distribution shifts.
>
> A key contribution of HLIP is to provide a framework that facilitates the development of robust models that are resilient to data noise and heterogeneity.
>
> **Report**
>
> As mentioned in W2, we leverage a large language model to reduce biases related to writing style, grammar, and wording habits.
>
> ---
>
> **references**
>
> [1] Todd Hollon, et al. Learning neuroimaging models from health system-scale data. Research Square, pp. rs–3, 2025.

---

> > ### Comment · Reviewer_NGWG · 2026-01-21
> > **Response to authors**
> >
> > Thank you for your clarifications and revisions to the manuscript, I consider my individual concerns catered to.

---

### Decision · Action_Editor_HXA2 · 2026-02-09

**Recommendation:** Accept as is

**Additional Comments:**

This paper introduces Hierarchical attention for Language-Image Pre-training (HLIP) for the effective and scalable pre-training of language-image models on 3D data. The core contribution lies in leveraging large-scale uncurated datasets through a three-level hierarchical attention mechanism (slice, scan, and study).

The paper initially received positive reviews; reviewers acknowledged the relevance of the lightweight attention scheme and the quality of the results for zero-shot transfer and linear probing. However, concerns were raised regarding the specifics of the attention protocol, preprocessing steps, the statistical rigor of the evaluation, and the limitations of using in-house data for reproducibility. The authors' rebuttal effectively addressed these points. Following the discussion period, all reviewers recommended acceptance.

The AE has carefully reviewed the submission and the discussion. The AE considers the attention mechanism to be relevant, as it enables large-scale pre-training for 3D medical imaging. The experimental results provide well-supported evidence for the claims. While broader data availability and the inclusion of error bars across all datasets would have further strengthened the paper, the AE considers the submission a valuable contribution to the community and therefore recommends acceptance.

**Audience:**

Yes

**Audience Explanation:**

This paper focuses on developing 3D foundation models for medical imaging, a subject of significant interest to the TMLR community.

**Claims And Evidence:**

Yes

**Claims Explanation:**

The paper introduces a lightweight attention mechanism for 3D medical images, enabling large-scale pre-training of language-image models. The approach is supported by experimental validation demonstrating strong performance in zero-shot transfer and linear probing.